# Chemical Structure, Optical and Dielectric Properties of PECVD SiCN Films Obtained from Novel Precursor

**Evgeniya Ermakova** [1,*]**, Konstantin Mogilnikov** [2]**, Igor Asanov** [3]**, Anastasiya Fedorenko** [3]**, Irina Yushina** [1]**, Vadim Kichay** [1]**, Eugene Maksimovskiy** [1] **and Marina Kosinova** [1,*]

1  Laboratory of Functional Films and Coatings, Nikolaev Institute of Inorganic Chemistry, SB RAS, Novosibirsk 630090, Russia
2  Laboratory of Physical Chemistry of Semiconductor Surfaces and Semiconductor-Dielectric Systems, Rzhanov Institute of Semiconductor Physics, SB RAS, Novosibirsk 630090, Russia
3  Laboratory of Nanomaterials, Nikolaev Institute of Inorganic Chemistry, SB RAS, Novosibirsk 630090, Russia
*  Correspondence: ermakova@niic.nsc.ru (E.E.); marina@niic.nsc.ru (M.K.)

**Abstract:** A phenyl derivative of hexamethyldisilazane—bis(trimethylsilyl)phenylamine—was first examined as a single-source precursor for SiCN film preparation by plasma enhanced chemical vapor deposition. The use of mild plasma (20 W) conditions allowed the preparation of highly hydrogenated polymeric-like films. The synthesis was carried out under an inert He atmosphere or under that of $NH_3$ with the deposition temperature range from 100 to 400 °C. The chemical bonding structure and elemental composition were characterized by Fourier-transform infrared spectroscopy, energy-dispersive X-ray analysis and X-ray photoelectron spectroscopy. The surface morphology was investigated by scanning electron microscopy. Ellipsometric porosimetry, a unique high-precision technique to investigate the porosity of thin films, was applied to examine the porosity of SiCN samples. The films were found to possess a morphologically homogenous dense defect-free structure with a porosity of 2–3 vol.%. SiCN films were studied in terms of their optical and dielectric properties. Depending on the deposition conditions the refractive index ranged from 1.53 to 1.78. The optical bandgap obtained using UV-Vis spectroscopy data varied from 2.7 eV for highly hydrogenated polymeric-like film to 4.7 eV for cross-linked nitrogen-rich film. The dielectric constant was found to decrease from 3.51 to 2.99 with the rise of hydrocarbon groups' content. The results obtained in this study were compared to the literature data to understand the influence of precursor design to the optical and electrical properties of the films.

**Keywords:** silicon carbonitride (SiCN) coatings; PECVD; thin film; single-source precursor; refractive index; optical bandgap; dielectric; hexamethyldisilazane

## 1. Introduction

PECVD using different single-source precursors is a convenient technique to create high-quality homogeneous coatings of various compositions [1]. The chemical bonding structure and composition of the films define their functional characteristics; it is most clearly expressed for non-stoichiometric compounds, like silicon carbonitride. Depending on the elemental composition, SiCN films possesses a set of properties that make them attractive as materials for different applications, such as different layers in modern semiconductor electronics [2–4], hard and protective coatings [5,6], piezoresistive material for pressure sensors [7], passivation and antireflective coatings for silicon solar cells [8], coatings in optoelectronics [9,10], etc.

In turn, the structure of the films depends on the design of the precursor and synthesis conditions. Organosilicon compounds of several classes were tested as precursors for SiCN film production. Among them, the most studied are disilazanes with a general formulae of $RN(SiMe_3)_2$, especially hexamethyldisilazane [11,12] and tetramethyldisilazane [13–15], due to its commercial availability, and hexamethylcyclotrisilazane [16],

bis(trimethylsilyl)ethylamine [17], 1,3-bis(dimethylsilyl)-2,2,4,4-tetramethylcyclodisilazane [16], N-methyl-aza-2.2.4-trimethylsilacyclopentane [18], and bis(tetramethylguanidine)dimethylsilane [19]. The group of aminosilanes $(R^1N)_x SiR^2_y$, including (dimethylamino)dimethylsilane [5], tris(dimethylamino)silane [5], bis(dimethylamino)methylsilane [20], methyltris(diethylamino)silane [21], tris(diethylamino)silane [22], and tris(dimethylamino)silane [23], were also studied. Two compounds of other classes, bis(trimethylsilyl)carbodiimide [24] and dimethyl(2,2-dimethylhydrazino)silane [25], were shown to be suitable as precursors for SiCN film production. Some general features of SiCN film formation can be distinguished. Films deposited from disilazane precursors are characterized by predominant Si–C binding, while the use of aminosilane precursors leads to the formation of layers rich in Si–N bonds. Use of carbodiimide as a precursor results in the formation of various CN bonds in the film. We have collected information about SiCN films obtained from different single-source precursors in the review [26]. Mild synthesis conditions (low deposition temperature and plasma power) allow incorporating the fragments from the initial molecule into the films structure with saving of the bonds presented in the fragment. It is mostly crucial for hydrogen-containing groups that define optical and dielectric properties of the material [27,28]. With an increase in the content of Si–C bonds (accompanied by a decrease in the concentration of Si–N bonds) and an increase in the carbon content in the film, the value of the refractive index increases while the transmittance and the band gap decrease [29]. Additionally, n decreases and $E_g$ increases with rise of the hydrogen content in the film in the form of C–H group, and the values differ with the type of organic group involved in the structure of the film [30]. Conversely, high hydrogen content, which participates in saturation of dangling bonds, leads to an increase in the values of the transmittance. The dielectric properties of SiCN films also depend strongly on film composition; the introduction of nitrogen into the film leads to an increase in the values of the permittivity, while high hydrogen content in organic groups is characteristic for low-*k* films [31,32]. Phenyl-containing polysilazanes are widely used as precursors for the synthesis of powder SiCN ceramics. However, in this case, a phenyl group does not remain in the film after annealing but tunes into the free carbon phase which affects the mechanical and electrical properties of the material [33]. Recently, studies have appeared devoted to the study of the effect of the phenyl group of the precursor on the structure and properties of silicon-based films. In [34] the films with the small pore size with narrow distribution were formed in benzene-bridged structure using $(EtO)_3Si$–$C_6H_4$–$Si(OEt)_3$. The films possessed improved mechanical and dielectric properties. In [35], $(C_6H_5)_2Si(N(CH_3)_2)_2$ was used as a precursor to obtain films with a porosity of 48% and a dielectric constant of 2.5. However, the integration of porous materials in ULIC technology is hampered by its low plasma-induced damage resistance, poor mechanical properties and susceptibility to stress-corrosion cracking. Previously, we have shown that SiC:H films deposited using trimethylphenylsilane as a precursor possessed properties of barrier layer in Cu interconnects [36]. The only phenyl-containing single-source precursor for SiCN film formation presented in literature is trimethylphenylaminosilane, which is related to aminosilanes. The use of this initial molecule allowed to produce the transparent films, but no other properties were studied [37].

In this work, bis(trimethylsilyl)phenylamine $PhN(SiMe_3)_2$ (BTMSPA) (Figure 1), which contains both a disilazane fragment and a phenyl group, was chosen as a single-source precursor. Previously we have shown that this compound is stable and possesses high vapor pressure at room temperature [38]. In this work we first examined it as a precursor in the CVD process. The SiCN:H thin films were deposited at low temperature by PECVD, and the correlation of optical and dielectric properties with structure and morphology of the films were investigated. The new data on porosity of the films were obtained by ellipsometric porosimetry.

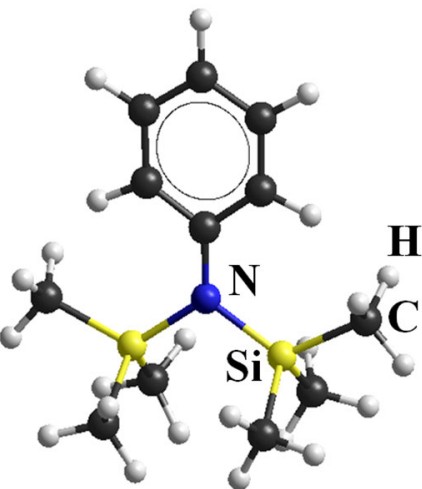

**Figure 1.** Bis(trimethylsilyl)phenylamine (BTMSPA).

## 2. Materials and Methods

### 2.1. Materials

The substrates used for SiCN film synthesis were the following: n-type Si(100) wafers with a thickness of 0.5 mm, Ge(111) wafers with a thickness of 0.2 mm, and fused silica slides of the dimensions $10 \times 5 \times 0.9$ mm. All substrates were preliminarily degreased by successive boiling in trichlorethylene and acetone. Then, an additional preparation procedure was performed depending on the type of substrate. Silicon wafers were treated in ammonia-peroxide hydrochloric-peroxide etchants. The native $SiO_2$ layer was removed by HF solution. Germanium wafers were treated by a solution of $HNO_3$, HF and $CH_3COOH$ (2:3:6). Each step of treatment was followed by washing in deionized water. Reagents of high purity grade were used. The substrates were dried in a nitrogen flow.

The single-source precursor for PECVD synthesis—bis(trimethylsilyl)phenylamine $PhN(SiMe_3)_2$—was synthesized using the method described in detail [38]. Helium and ammonia of high purity grade were used as plasma-forming gases.

### 2.2. PECVD

The SiCN films were deposited by the PECVD process. The scheme of the in-house-built CVD equipment is presented in [39]. The tunnel-type quartz reactor includes a plasma-generating section coupled with a radiofrequency (RF) plasma source (40.68 MHz) and film-forming zone with substrates placed inside. The deposition zone is equipped with a heating system which varied the synthesis temperature from 100 to 450 °C. The reactor has two inputs: one for plasma-forming gases, one for vapors of liquid precursor BTMSPA. Helium and ammonia were used as plasma-forming gases. Before the experiment the system was evacuated to a pressure of $4 \times 10^{-3}$ Torr. The pressure of BTMSPA was kept at $1.5 \times 10^{-2}$ Torr; the plasma-forming gas was at $6 \times 10^{-3}$ Torr and the power used was 20 W for all experiments based on the preliminary optimization of the PECVD process.

### 2.3. Characterization of Chemical Bonding Structure and Surface Morphology

Fourier Transform Infra-Red (FTIR) spectra of SiCN/Si(100) samples were recorded at room temperature using FTIR-spectrophotometer SCIMITAR FTS 2000 (Digilab) in the range of 400–400 $cm^{-1}$ with a resolution of 2 $cm^{-1}$. Each spectrum was normalized by the thickness of the film in order to compare the bond intensities.

XPS spectra of SiCN films were measured on a FlexPS spectrometer with a PHOIBOS 150 analyzer (Specs GmbH, Berlin, Germany) using a non-monochromatic Al K$\alpha$ excitation radiation (1486.61 eV). The Casa XPS software (Casa Software Ltd., Teignmouth, UK) was used for data processing. The C1s, N1s, and O1s spectra were fitted by mixed Gaussian-Lorentzian peaks after subtraction of a Shirley background. Preliminary calibration of

binding energy (BE) scales of XPS spectrometer is also carried out using the Au 4f7/2, Ag 3d5/2, Cu 2p3/2 and Cu L3VV lines to test for the BE scale linearity, confirm the uncertainty of the scale calibration, correct for small drifts of that scale and define the expanded uncertainty of the calibration of the BE scale. The BE were then calibrated to the C-C component energy of C1s peak at 284.6 eV. The calibration method choice was determined by the need to consider the shift of the lines due to the charging of the samples.

Scanning Electron Microscopy was used to investigate the surface morphology of the SiCN/Si(100) samples. High-resolution images of the surface and cross-sectional view were obtained using a field-emission scanning electron microscope JEOL JSM 6700F (Jeol, Tokyo, Japan) with the accelerated voltage of 5 keV in secondary electron mode. The microscope is equipped with an Energy-Dispersive X-Ray (EDX) analyzer Quantax 200 (Bruker, Berlin, Germany) that was used for elemental analysis of SiCN/Ge samples.

### 2.4. Optical Properties

The film thickness and refractive index of SiCN films deposited on Si(100) wafers were determined using monochromic null ellipsometry at a wavelength of 632.8 nm (He-Ne laser). The deposition rate was calculated by dividing the ellipsometric film thickness to synthesis time. The values of the refractive index were calculated with an error of 0.03, which is determined by measuring the parameters $\psi$ and $\delta$, as well as the initial model choice of the inverse problem.

Transmittance of the films deposited on $SiO_2$ substrates was determined using Shimadzu UV-3101PC scanning spectrophotometer in the range of 200–2000 nm with a resolution of 2 nm. The transmittance curves were used to estimate data on optical bandgap using the Tauc model [40].

### 2.5. Porosity

The porosity of the films was studied using atmospheric pressure ellipsometric porosimetry. The basic principles of this method are presented in [41]. The ellipsometric parameters $\psi$ and $\delta$ were determined with the use of a spectroscopic ellipsometer ELLIPS 1891 and then turned to optical constants ($n(\lambda)$, $k(\lambda)$, $d$) using a model of isotropic and homogeneous single-layer transparent film deposited on a smooth substrate surface. Adsorption tests were performed to investigate the porosity of the films. Isopropyl alcohol (IPA) vapors were chosen as an adsorptive. The porosity of the films is calculated as the volume of adsorbed IPA from the values of refractive indices (RI) measured during the test.

### 2.6. Dielectric Constant

The Metal-Insulator-Semiconductor (MIS) structures were prepared by Al electrodes with an area of 0.5 $mm^2$ formation onto the SiCN/Si(100) structures by Al target sputtering using shadow mask. The Capacitance-Voltage (CV) measurements of MIS structures were performed in a quasi-stationary mode at room temperature. The error in determining the permittivity is less than 5% and is a consequence of the mismatch between the contact area during the deposition of metallization in the process of creating the MIS structure, and the error in calculating the thickness of the SiCN films.

## 3. Results and Discussion

### 3.1. Chemical Bonding Structure and Elemental Composition

The chemical bonding structure of SiCN films prepared at different deposition conditions was characterized by FTIR and XPS analysis. Figure 2 presents the evolution of FTIR spectra of SiCN films deposited using BTMSPA/He (Figure 2a,b) and BTMSPA/NH₃ (Figure 2c,d) gas mixtures. The assignment of the absorption bands was performed using the literature data on SiCN film investigation [12,42–44]. The spectra include a main broad absorption band in the "fingerprint region" at 700–1300 $cm^{-1}$, peaks with a maximum position at 1260, 1450, and 2750–3000 $cm^{-1}$ attributed to $\delta_s$(CH), $\delta_a$(CH) and $\nu_a$(CH)+$\nu_s$(CH) bond vibrations, features at 1595 and 3300–3400 $cm^{-1}$ corresponded to $\delta$(NH) and $\nu$(NH)

vibrations, and a weak peak at 2180 that can be assigned to $\nu$(C$\equiv$N) or $\nu$(Si–H) vibration. The full description of BTMSPA FTIR spectrum is presented in [38]. It should be noted that it contains a wide absorption band in the range of 3000–3200 cm$^{-1}$, corresponding to C–H vibrations in phenyl group. It can be assumed that the spectra of PECVD films should also contain this absorption band, but it is not observed. There are several reasons for the absence of this absorption band. First, the saturation of the phenyl to cyclohexane group or even the opening of the benzene ring with the formation of long aliphatic chains is possible. Indeed, it was demonstrated in the literature that the polymerization of benzene in an RF glow discharge leads to the formation of a film whose IR spectrum contains intense bands in the region of 2800–3000 cm$^{-1}$, corresponding to vibrations of the C–H bond in aliphatic groups, while the band at 3200 cm$^{-1}$ has low intensity [45], or is not detected at all [46]. The source of additional hydrogen atoms required to saturate this fragment can be methyl groups contained in the disilazane part of the monomer. The appearance of atomic hydrogen in the plasma of different organosilicon compounds was shown earlier by the OES [43,47,48]. On the other hand, the elimination of the phenyl fragment and its removal from the deposition zone by the gas flow cannot be excluded. The IR features correspond to N–H vibrations and were detected in all spectra of the films deposited using BTMSPA/He mixture. The only source of nitrogen in this case is the monomer molecule therefore, the formation of the N–H bond occurs with the elimination of the phenyl group or the –Si(CH$_3$)$_3$ group fragmentation, which in both cases is associated with a decrease of C–H band intensities observed in all ranges of deposition temperatures.

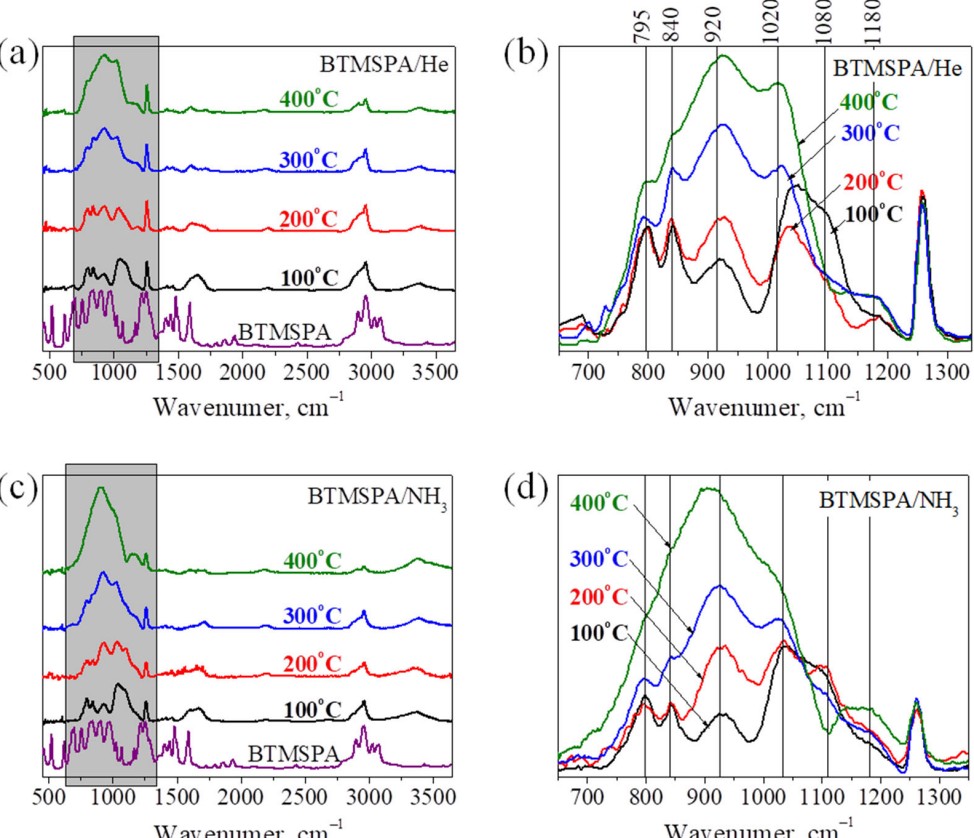

**Figure 2.** Evolution of wide-range FTIR spectra of SiCN films deposited using BTMSPA/He (**a**) or BTMSPA/NH$_3$ (**c**) mixtures, and magnification of selected area: (**b**) for BTMSPA/He, (**d**) for BTMSPA/NH$_3$ gas mixtures.

The range of 700–1300 cm$^{-1}$ (Figure 2b) includes the most important absorption bands that characterize the film formation process and at least six Gaussian components are necessary for their description. They consist of overlapping components centered

at 795, 840, 920, 1020, 1080 and 1180 cm$^{-1}$, corresponding to $\nu$(Si–C), $\rho$(C–H), $\nu$(Si–N), $\omega$(Si–CH$_x$–Si), $\nu$(Si–O) and $\nu$(C–N) bands. The main trend observed in this region with rise of film deposition temperature is an increase in the intensity of the Si-N component with a simultaneous decrease in the contribution of the C-H peak. This change in the nature of the spectra indicates the increasing importance of the processes of hydrocarbon fragments cleavage with the formation of a cross-linked structure, during the plasma deposition. For the SiCN films deposited using the BTMSPA/NH$_3$ mixture (Figure 2d), the significant increase in the contribution of the absorption band at 917 cm$^{-1}$ corresponding to Si–N stretching vibrations is observed due to the excess of nitrogen-containing species in the gas phase. This high-intensity Si-N band absorption indicates the formation of 'silicon nitride-like' skeleton.

The tendency of deposited films toward partial oxidation should be noted. This is evident by the presence of absorption bands in the 1080 and 1700 cm$^{-1}$ region, corresponding to the $\nu$(Si–O) and $\nu$(C=O) vibrations. Films prepared at a temperature of 100–200 °C undergo the greatest changes, while the further increase in the deposition temperature resulted in disappearance of absorption bands attributed to oxygen-containing bonds. At the same time, a decrease in the intensity of peaks characteristic for hydrogen-containing bonds and increase in Si-N band intensities are observed; therefore, better stability of the film may be associated with the increasing crosslinking in the film and the transition from a 'polymeric-like' to an inorganic chemical structure.

The local bonding environment was additionally studied by XPS. Figure 3 represents the full-range spectra of SiCN films deposited using He (Figure 3a) or NH$_3$ (Figure 3c) as plasma-forming gas at the deposition temperature of 400 °C. All spectra contained Si-, C-, N-, and O-related components, and no other peaks caused by contamination were observed. There is a significant change in the XPS spectra with a variation of the type of additional gas, expressed in a significant decrease in the C1s component intensity along with an increase in N1s peak.

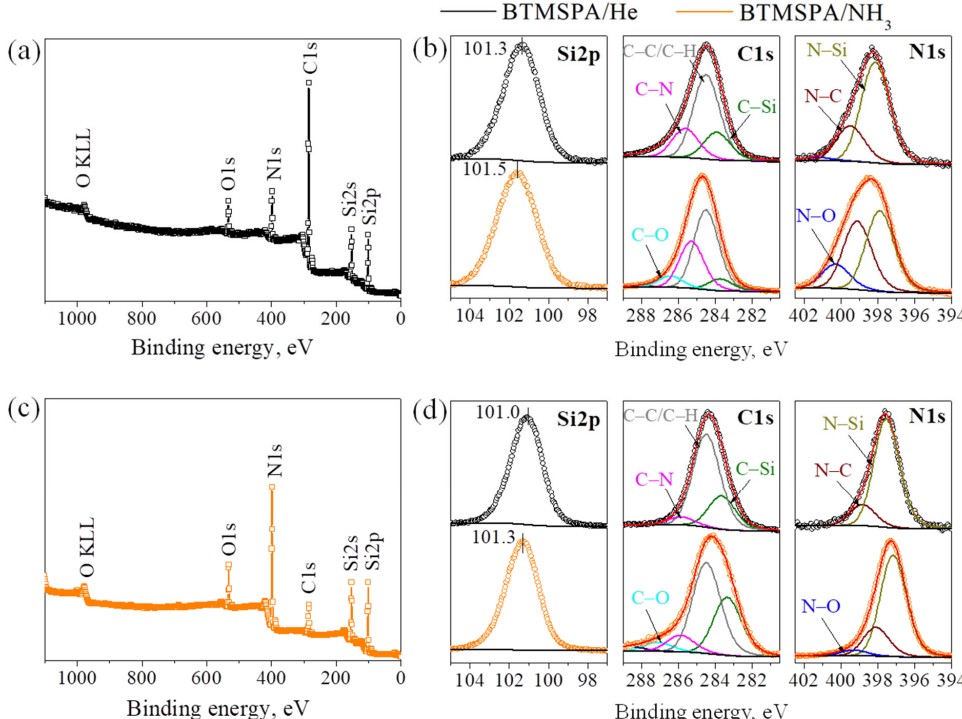

**Figure 3.** Full-range XPS spectra of SiCN films deposited at 400 °C using BTMSPA/He (**a**) or BTMSPA/NH$_3$ (**c**) mixtures, and Si2p, C1s, N1s core-level spectra of SiCN films deposited at 100 °C (**b**) and 400 °C (**d**) using BTMSPA/He (black) or BTMSPA/NH$_3$ (orange) mixtures.

Si2p, C1s and N1s core-level spectra are presented in Figure 3b,d. The spectra interpretation was performed using the literature data [19,44,49,50]. The width of the Si2p peak exceeds 2 eV, which can be interpreted as a superposition of at least two separate components of Si–C and Si–N; however, the fitting is problematic. A maximum of the peak is in the area of 101.0–101.5 eV, which is an intermediate position between the values characteristic of Si-C and Si-N bonds in silicon carbide and silicon nitride, respectively. With an increase in the deposition temperature and the introduction of ammonia, the maximum shifts slightly to the region of high binding energies, that can be associated both with an increase in the contribution of the Si–N component and the appearance of oxygen in the silicon environment. The C1s peak can be fitted with 4 main components at 283.8, 284.6, 285.4, and 286.5 eV, that can be attributed to C–Si, C–C/C–H, C–N, and C–O bonds, correspondingly. The relative intensity of the C–N component increases with the replacement of inert helium with ammonia. The contribution of the Si–C component, on the contrary, decreases. The data are consistent with the results of IR analysis. With an increase in the synthesis temperature for both gas systems a decrease in the relative contribution of the C–N component is noted. The results agree with data on the decomposition of the N1s peak. Three main components can be distinguished in the regions of about 397.2, 398.2, 399.4 eV, corresponding to the N–Si, N–C and N–O bonds. With an increase in the synthesis temperature and/or the use of ammonia as a plasma-forming gas, an increase in the relative intensity of the N-Si component is observed. The presence of the N–O component is typical for SiCN films deposited using ammonia as additional gas. This indicates the oxidation of NH groups is one of the most probable aging mechanisms of films [43].

Thus, XPS analysis show that low-temperature films contain predominantly Si–C, Si–N, C–N, and C–C bonds. The presence of the C–N bond in the BTMSPA/He system indicates the incomplete destruction of the precursor molecule in the plasma. Considering the IR analysis data, which show the absence of absorption bands related to C–H (aromatic) bonds, it can be assumed that the phenyl fragment is saturated by hydrogen during film formation. With the introduction of ammonia, a significant increase in the contribution of the C–N bond is observed, indicating the formation of new bonds occurring as a result of chemical reactions involving the products of decomposition of ammonia in the plasma. In addition, the films obtained in the ammonia gas system contain an N–O bond on the film surface, which probably appears when the N–H bond is oxidized in the environment after the synthesis. With an increase in the synthesis temperature, a decrease in the relative contribution of the C–N bond is observed; thus, the chemical structure of the films includes mainly Si–C and Si–N bonds.

Elemental composition of SiCN films is presented in Figure 4. The data on the additional sample deposited at 450 °C was added to confirm the tendencies of composition changes. The element concentrations in the film bulk were determined by EDX that do not provide the information on hydrogen concentration. The evolution of hydrogen was discussed above, considering the FTIR spectroscopy data on C–H and N–H bands. EDX revealed that all the films contain silicon, carbon, nitrogen and oxygen in their composition. The C/Si ratio in the films deposited at low temperature exceeds the C/Si ratio in the precursor molecule significantly. The rise of deposition temperature resulted in a drastic decrease in the C/Si ratio and identify the thermally induced elimination of organic groups. Considering the FTIR spectroscopy data it can be proposed that decomposition includes elimination of methyl-containing groups. One possible path is exclusion of –SiMe$_x$ moieties as a result of weakening of N–Si bonds caused by a pull of electron density from nitrogen by a phenyl substituent. Since one N–Si breakage leads to remove of up to three Si–C bonds involved in –SiMe$_3$ group, this assumption does not contradict the FTIR data. Ammonia addition into the reactive mixture resulted in a decrease of the C/Si ratio due to elimination of carbon-containing fragments stimulated by presence of active gas species in plasma, such as H and NH [51]. Oxygen concentration in the films varied in the range of 0–10 at.% and decreased with rise of deposition temperature. At the same time, this value was higher in the SiCN films deposited from the BTMSPA/NH$_3$ mixture. The oxygen atoms are not

presented in the precursor and additional gases molecules but may originate from several sources. The most possible reason is oxidation or hydroxylation of dangling bonds, formed in plasma process, during the storage in air. However, the impact of etching of the walls of the quartz reactor by plasma species and residual air in vacuum cannot be excluded.

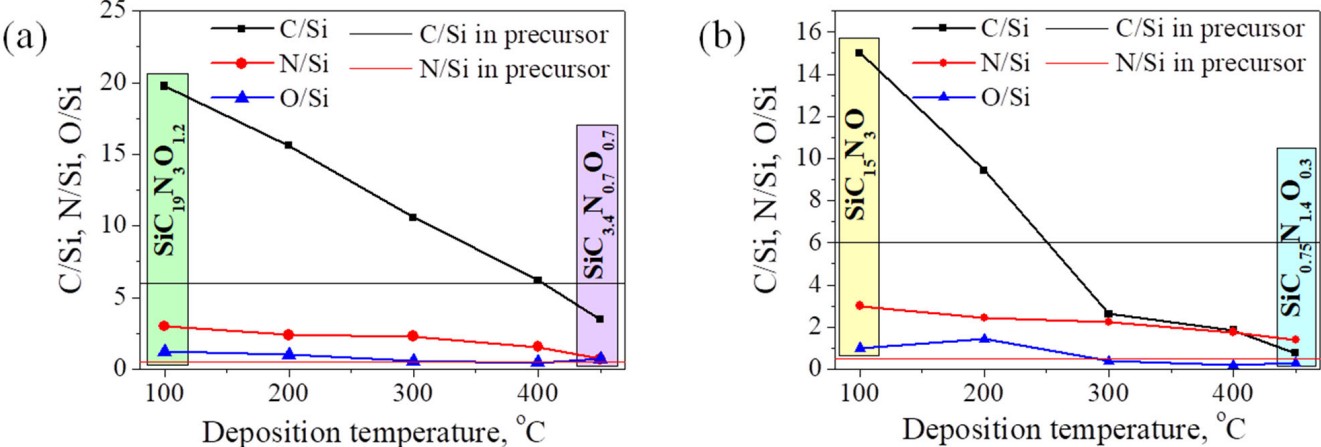

**Figure 4.** Elemental composition of SiCN films deposited using BTMSPA/He (**a**) or BTMSPA/NH$_3$ (**b**) mixtures.

### 3.2. Microstructure and Porosity

Figure 5 presents SEM typical images for the SiCN film surface and a cross-section view. According to these images, all the films are uniform, dense, homogeneous, and have a microstructure without pores and cracks. No crystalline form was observed, indicating their amorphous state.

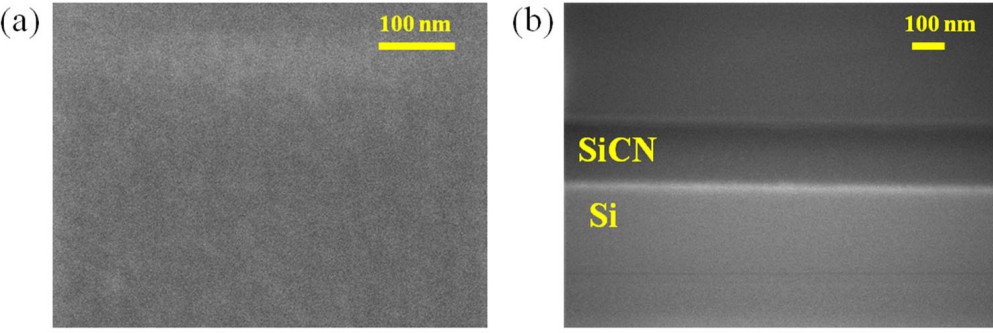

**Figure 5.** SEM images of SiCN films: surface (**a**) and cross-section view (**b**).

The pore structure was investigated by ellipsometric porosimetry. This technique was proven to be the effective way to determine the pore volume in the film materials [52]. The porosimetry of the films was studied by adsorption of IPA (isopropyl alcohol) molecules. Figure 6 presents the adsorption-desorption isotherms of SiCN films deposited using the BTMSPA/He mixture, deposited at 100 and 400 °C. The presence of a hysteresis loop on the adsorption-desorption isotherm can be interpreted as the formation of pores with narrow channels near 5 nm in size, that are difficultly available for IPA molecule transport. The total pore volume of the films was 3 and 2 vol.% for the films obtained at 100 and 400 °C, respectively.

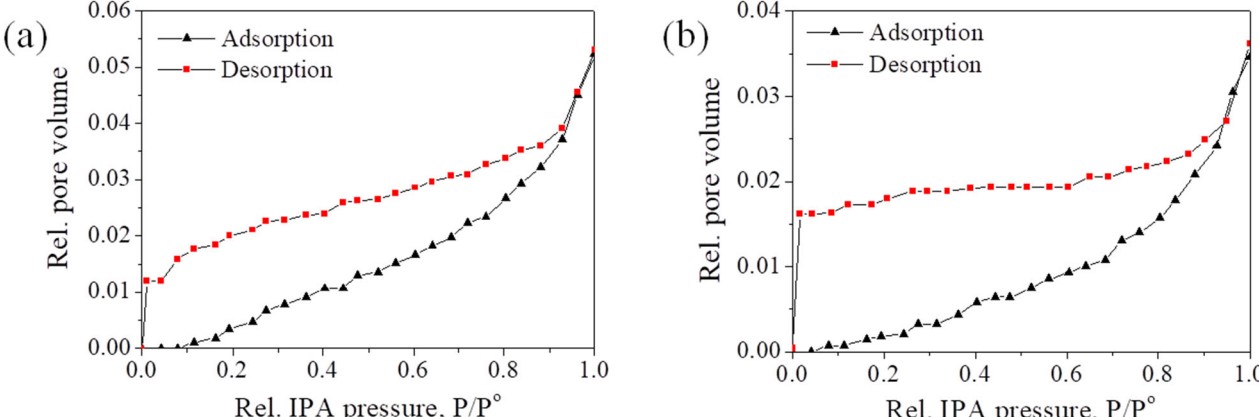

**Figure 6.** Adsorption and desorption curves of isopropyl alcohol (IPA) on the surface of SiCN films deposited using BTMSPA/He mixture at 100 (**a**) and 400 °C (**b**).

The combination of SEM and porosimetry data point out the homogeneous dense structure of the films on the macro- and microlevel.

### 3.3. Optical Properties

The refractive index of the films was measured by laser ellipsometry at 632.8 nm (Figure 7).

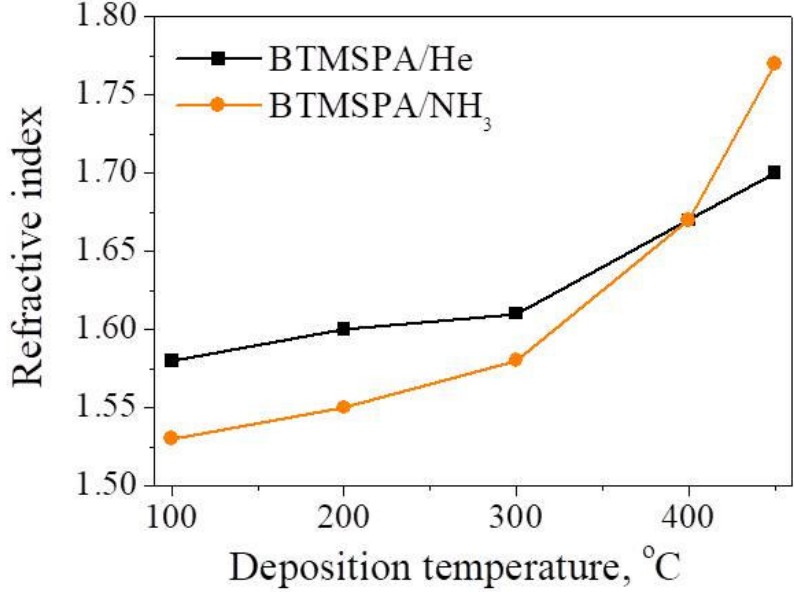

**Figure 7.** Evolution of refractive index of SiCN films deposited using different reactive mixtures.

The *n* values for films deposited at 100–200 °C are 1.53–1.58 which is typical for SiCN:H material [14,21,27]. The increase in the refractive index with rising of synthesis temperature is observed in SiCN films deposited using both BTMSPA/He and BTMSPA/NH$_3$ mixtures. This tendency is determined by two combining factors. First, the temperature-stimulated elimination of hydrocarbon fragments occurs, and hence the hydrogen content decreases, which was demonstrated by FTIR spectroscopy. The strong dependence of the refractive index on the hydrogen content was previously shown for SiCN:H and SiN:H films obtained by both CVD and PVD techniques [30,53]. On the other hand, an increase in the Si/N ratio is observed which means that the film composition changes from SiCN-like to SiN-like. In turn, it is known that the refractive indices of single-crystal Si$_3$N$_4$ and nitrogen-rich SiN$_x$ films are 1.97 and 1.91–2.04, respectively [54]. It was shown by elemental analysis that the

N/Si ratio was 0.7 and 1.37 for the SiCN films produced at 450 °C using BTMSPA/He and BTMSPA/NH$_3$ mixtures, respectively. The deeper transition of chemical bonding structure and composition of the film to silicon nitride in the case of the SiCN film obtained from BTMSPA/NH$_3$ mixture explains the higher values of the refractive index

　　　Figure 8a,c presents UV-Vis spectra of the SiCN/SiO$_2$ structures, and SiO$_2$ substrate for comparison. All films were about 200 nm in thickness. Low interference fringes can be noticed in the spectra due to the interference at the interfaces between air, film, and SiO$_2$ substrate.

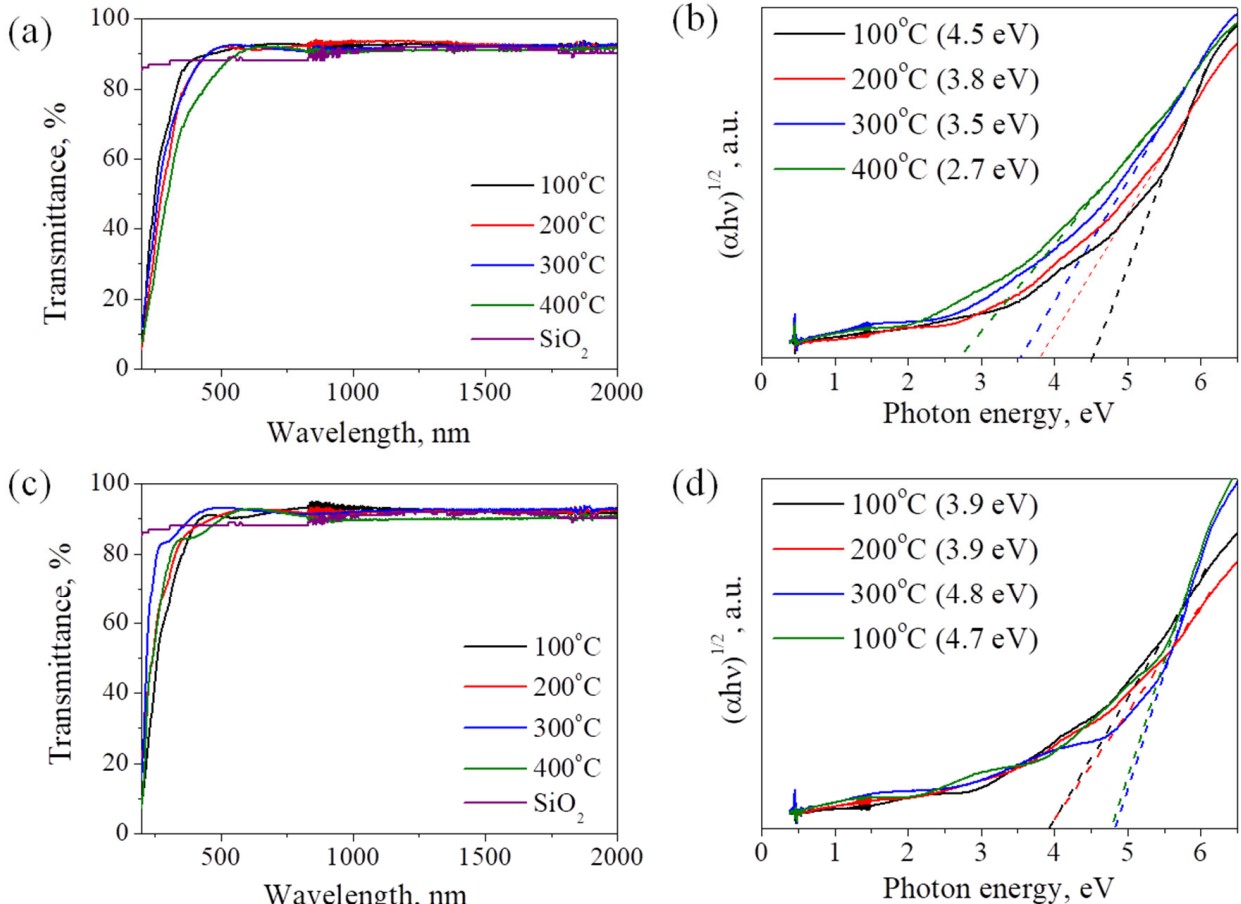

**Figure 8.** Transmittance and absorbance spectra in Tauc plots of SiCN films deposited using BTMSPA/He (**a**,**b**) and BTMSPA/NH$_3$ (**c**,**d**) mixtures.

　　　The samples possessed the transmittance higher than SiO$_2$ at wavelength of 520 nm and more and resulted in a decrease of reflectance in comparison with fused silica. At the shorter wavenumbers, a sharp decrease of transmittance can be seen that originated from the fundamental absorption of the films. For the SiCN films deposited using the BTMSPA/He mixture, the absorption band exhibits the redshift with an increase of deposition temperature and is caused by their strong crosslinking. Ammonia addition has no clear effect on transmittance due to several parameters, which are discussed below. The bandgap of the films was calculated using the Tauc equation:

$$\alpha h\nu = B(h\nu - E_g)^n \tag{1}$$

where $\alpha$ is absorption coefficient, $h\nu$ is the incident photon energy, n is the parameter related to the density of states distribution, and B includes information on the convolution of valence band and conduction band, as well as degree of topological disorder in the film [40,55–58]. $E_g$ was estimated assuming n = 2 [5] by the extrapolation of linear part

of the plot $(\alpha h\nu)^{1/2} - h\nu$. The error of the measured bandgap, defined by standard errors of the slope and intercept of linear fit determine, for all samples did not exceed 0.1 eV. The obtained $E_g$ values (Figure 8b,d) for SiCN films deposited using BTMSPA/He mixture decreased with deposition temperature from 4.5 to 2.7 eV. For films deposited using the BTMSPA/NH$_3$ mixture, the optical bandgap remains constant in two temperature regimes. The low-temperature samples that were polymeric-like films with high content of hydrocarbon fragments possess the bandgap of 3.9 eV, which is close to low-temperature films produced with helium. The stepwise increase of $E_g$ to 4.7–4.8 eV with the rising deposition temperature may be associated with transformation from carbon-rich SiCN to nitrogen-rich SiCN. Indeed, this value is close to bandgap of Si$_3$N$_4$, that is 5.2 for single crystal and 4.8 for amorphous SiN$_x$:H [59].

### 3.4. Dielectric Properties

The dielectric constant was derived from capacitance-voltage (C–V) measurements at room temperature on Si(100)/SiCN/Al structures. Figure 9 shows the temperature dependence of the resulted dielectric constant. The SiCN films deposited using BTMSPA/He gas mixture at 100–400 °C possessed $k$ values in the range of 2.99–3.51 due to the high CH$_x$ group concentration. Permittivity of samples produced from the BTMSPA/NH$_3$ source was 3.73–3.77, which is higher and increased up to 5.39 with rising deposition temperature due to the elevated nitrogen concentration in the film.

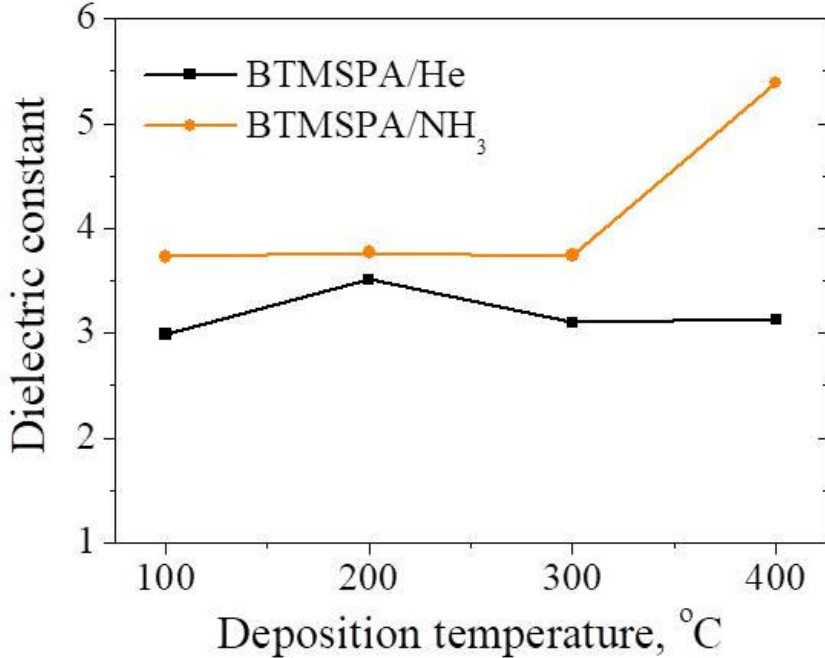

**Figure 9.** Temperature dependences of dielectric constant of SiCN films deposited using BTMSPA/He and BTMSPA/NH$_3$ mixtures.

The data presented make it possible to identify the main features in the formation of SiCN layers using BTMSPA as a precursor, as well as the relationship between the synthesis conditions, structure, elemental composition, and properties of the films. The initial compound, BTMSPA, undergoes significant changes under low power plasma conditions both in an inert atmosphere of helium and in a mixture with ammonia. The most interesting changes concern the phenyl group. The absence of absorption bands corresponding to C–H (sp$^2$) vibrations in all FTIR spectra of the films indicate that it is not included in the film. In turn, the presence of the C–N and N–C components in the XPS C1s and N1s spectra confirm that the phenyl group is not eliminated, but saturated by hydrogen contained in plasma. This is also evident by the presence of the C–H absorption band (840 cm$^{-1}$), which is characteristic of vibrations in long aliphatic chains. On the contrary, the introduction of

ammonia into the reactive mixture results in a change of the spectra of the films, including a drop of both the relative intensity of C–N components in the XPS spectra and absorption bands corresponding to C–H vibrations in the FTIR spectra. This indicates that with an increase in the concentration of energetically active particles (including hydrogen atoms) in the gas phase, formed during the activation of ammonia, a deeper decomposition of the initial substance and/or bombardment of the surface of the growing film occur. As a result, hydrocarbon fragments are eliminated. An increase in temperature in the case of both gas mixtures additionally stimulates the splitting off of hydrogen-containing fragments with the formation of a cross-linked structure, which is typical for silicon carbonitride films and was described in the literature [60]. As a result of these processes, the Si–N bond becomes the main bond in the films, which is mostly pronounced for the $BTMSPA/NH_3$ mixture.

All these changes significantly affect the properties of the deposited films. Due to the Lorentz-Lorenz correlation, the films' refractive index is associated with the polarization response of a material that depends on the chemical bonding structure, and density of the layer [61,62]. The rise of refractive index with deposition temperature observed for films deposited using both gas mixtures is related to the increasing density of the film. This effect for SiCN films was investigated in detail by Prof. A.M. Wrobel's group [5]. The authors explain the effect by the thermally induced crosslinking reactions resulting in the formation of the Si$-$C$-$N network. On the other hand, the films produced using the $BTMSPA/NH_3$ gas mixture are characterized by higher nitrogen content and Si–N bond concentration that is confirmed by FTIR, XPS and EDX analysis, and as a result, in the lower values of $n$. With an increase of synthesis temperature, the refractive index grows faster than in the $BTMSPA/He$ mixture. We assume the reason is that the densification of the films occurs rapidly due to the additional bombardment of the growing film by active species.

As for the optical bandgap, the $E_g$ values are mainly defined by chemical structure and composition of the films. The band gap is significantly affected by the presence of hydrogen-containing bonds, especially Si–H bonds. [28] According to FTIR spectra analysis, the intensity of this absorption band decreases for both gas mixtures with the rise of synthesis temperature, which contributes to a drop in $E_g$ values. The films produced using the $BTMSPA/NH_3$ mixture are characterized by a higher N/Si ratio and higher relative concentration of Si–N bond in FTIR and XPS spectra, thus their bandgap increases. The value of 4.8 eV is close to the bandgap of the films, deposited with the excess of ammonia in reactive mixture, that can reach 5.4 eV [63]. It should be emphasized that a stepwise change in the value of $E_g$ with an increase in the nitrogen content in the films was found in several works [64–66]. The authors attribute such a change to the fact that even with a small addition of a nitrogen-containing reagent, due to the more preferential formation of Si-N bonds, the composition of the films changes sharply and becomes close to silicon nitride.

### 3.5. Comparison with the Literature Data

The precursor molecule, BTMSPA, can be regarded, on the one hand, as a member of disilazane family and, on the other hand, as a phenyl-containing compound. The results presented in this work are compared with the properties of the films synthesized using other precursors related to these groups (Table 1). It is known that the process conditions have a significant effect on the composition, structure, and properties of the resulting films. This table summarizes the data for the films obtained in mild deposition conditions by the PECVD technique with the close flow rates of the precursor and plasma-forming gas. The characteristics collected are for the compounds of disilazane row: hexamethyldisilazane $HN(SiMe_3)_2$ (HMDSN), bis(trimethyl)ethylamine $EtN(SiMe_3)_2$ (BTMSEA), and phenyl-containing compounds: trimethylphenylsilane $PhSiMe_3$ (TMPS) and trimethylphenylaminosilane $PhHNSiMe_3$ (TMPAS).

**Table 1.** Properties of the SiCN films deposited from various organosilicon precursors in a mixture with He or NH$_3$ using PECVD (data reported in the literature and present study).

| Property | Precursor (*add. gas*) | | | | |
|---|---|---|---|---|---|
| | Phenyl-Containing Precursors | | | Disilazane-Based Precursors | |
| | BTMSPA | TMPS | TMPAS | HMDSN | BTMSEA |
| Refractive index | 1.58–1.68 (*He*) 1.53–1.67 (*NH$_3$*) | 1.62–1.69 (*He*) 1.53–1.63 (*NH$_3$*) | 1.5–1.65 (*He*) 1.32–1.65 (*NH$_3$*) | 1.5–1.80 (*He*) 1.52–1.73 (*NH$_3$*) | 1.54–1.74 (*He*) 1.47–1.56 (*NH$_3$*) |
| Optical bandgap | 2.7–4.5 (*He*) 3.9–4.7 (*NH$_3$*) | 2.7–3.7 (*He*) 2.48–4.03 (*NH$_3$*) | 2.55–3.5 (*He*) 3.55 (*NH$_3$*) | 3.2–4.3 (*He*) 2.2 (*NH$_3$*) | 4.15–4.37 (*He*) 3.9–5.12 (*NH$_3$*) |
| Permittivity | 2.99–3.51 (*He*) 3.73–5.39 (*NH$_3$*) | 2.96–3.73 (*He*) 3.36–5.29 (*NH$_3$*) | - | 3–4.2 (*He*) 4.7–6.2 (*NH$_3$*) | >5.8 |
| Ref. | This paper | [36] | [5,37] | [11,16,67] | [17] |

The refractive indices of low-temperature polymer-like films obtained in an inert atmosphere are similar for all selected precursors. In the case of the disilazane compounds, there is a slight increase in the *n* values in the row HMDSN–BTMSEA–BTMSPA. The difference in the structure of the films obtained from these precursors consists mainly in the length of the aliphatic bridges, which are affected by the size of the substituent. With the rise of the deposition temperature, an increase in the refractive index is observed for the films deposited using all precursors. This tendency is caused by a change in the composition and bonding environment in the films due to the elimination of hydrocarbon fragments and, consequently, an increase in cross-linking and a decrease in the hydrogen content. In this case, in the disilazane series, a correlation is observed between the refractive index and the N/C ratio. Indeed, with an increase in the nitrogen content in the precursor, there is an increase in the nitrogen concentration in the films and a transition to 'silicon nitride'-like films (refractive index of Si$_3$N$_4$ is 1.97). For the phenyl-containing precursors, such a strict dependence is not observed due to the presence of big hydrocarbon fragments that determine the properties of the film. The type of precursor chosen also affects the band gap of the resulting material. Films synthesized from phenyl-containing precursors have band gaps of 2.5–3.5 eV. The use of disilazane precursors (HMDSN, BTMSPA, and BTMSEA) significantly enlarges $E_g$ up to 4 eV and more. As well as the band gap, the permittivity of the films, synthesized using phenyl-containing precursors was lower. The *k* values of the SiCN films were 2.96–3.70, which makes them promising for replacing porous low-*k* dielectrics, when the technological process includes plasma induced damage of porous materials.

## 4. Conclusions

Novel organosilicon precursor bis(trimethylsilyl)phenylamine PhN(SiMe$_3$)$_2$ (BTMSPA), which is a phenyl-derivative of hexamethyldisilazane, was first examined as precursor for SiCN film synthesis. The aim of this work was to investigate the structural features, optical (refractive index, transmittance, optical bandgap) and dielectric properties of the films produced using BTMSPA and compare the results with data on relative precursors reported in the literature. The films were produced by low-power PECVD in the temperature range of 100–400 °C using different plasma-forming gases (He or NH$_3$) that have different reactivity in glow-discharge conditions. The chemical bonding structure and elemental composition analysis show that the deposition temperature played a crucial role in film formation. The thermally activated elimination of –SiCH$_3$ groups following crosslinking took place. The main bonds defined for low-temperature films were Si–C, Si–N, C–N, C–H, and C–C. The phenyl group was not eliminated but was saturated by hydrogen atoms with formation of aliphatic bridges. The increase of synthesis temperature resulted in the deposition of films with domination of the Si–N bond that is more pronounced for the BTMSPA/NH$_3$ gas mixture. The SEM and ellipsometry porosimetry analysis show that the films possess morphologically homogenous dense defect-free structures with a porosity of

2–3 vol.%. Depending on the deposition conditions, the refractive index ranged from 1.53 to 1.78. The optical bandgap obtained using UV-Vis spectroscopy data varied from 2.7 eV for highly-hydrogenated polymeric-like film to 4.7 eV for cross-linked nitrogen-rich film. The dielectric constant was found to decrease from 3.51 to 2.99 with rise of the hydrocarbon groups' content. A comparison with the literature data show that the selection of the precursor for SiCN film deposition is an important step that determines characteristics of the resulted films. The use of the disilazane precursors allows gently influencing the optical and dielectric properties of the films by changing the type of N-substituent. On the contrary, the films deposited using all phenyl-containing compounds possesses nearly constant optical and dielectric properties, which indicate the interchangeability of these precursors.

**Author Contributions:** Conceptualization, E.E. and M.K.; methodology, E.E. and M.K.; formal analysis, I.Y., V.K., E.M. and I.A.; investigation—E.E., K.M. and A.F.; writing—original draft preparation, E.E.; writing—review and editing, M.K.; supervision, M.K.; project administration, M.K. All authors contributed to the discussion and interpretation of the results. All authors have read and agreed to the published version of the manuscript.

**Funding:** The research was supported by the Ministry of Science and Higher Education of the Russian Federation, project N 121031700314-5.

**Institutional Review Board Statement:** Not applicable.

**Informed Consent Statement:** Not applicable.

**Data Availability Statement:** Not applicable.

**Acknowledgments:** The authors thank N.I. Alferova for the FTIR spectroscopy characterization.

**Conflicts of Interest:** The authors declare no conflict of interest.

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
