# Peer review of "Chemical Structure, Optical and Dielectric Properties of PECVD SiCN Films Obtained from Novel Precursor"

_coatings, doi:10.3390/coatings12111767_

Round 1

Reviewer 1 Report

The paper is based on the structural, optical and dialectical properties PECVD films obtained from novel precursor and characterized with the techniques like FTIR, EDAX, XPS, SEM, Ellipsometry, UV-Vis measurements. The authors have found the decrease of dielectric constant with increase of hydrocarbon groups and also change in refractive index along with bandgap energy. The authors have corelated the observed parameters with each other, however, still there are some more discussions are required for the same. There are some major points needs to be addressed with the suggested references. So, my recommendation is Major Revision of the manuscript.

Comments:

1.      Novelty of the work should be highlighted in the introduction in more clearly.

2.      The authors designed the work nicely, merely presented the results but failed to discuss the observed results elaborately.

3.      The SEM image are not marked with the scale. If possible, give better resolution images.

4.      Why the refractive index increased with synthesis temperature? Any reason for the lower value of refractive index for BTMSPA/He at 450 Degree Celsius than BTMSPA/NH3.

5.      Fig.7b, 7d is not seen clearly. Give a better plot for this.

6.      What is the significance of parameter B in equation 1. Refer- Journal of Non-Crystalline Solids 355 (37-42), 1943-1946 (2009), Journal of Applied Physics 104 (5), 053501 (2008).

7.      The bandgap values should be mentioned with error values.

Reviewer 2 Report

This paper reports a novel organosilicon precursor for the SiCN film synthesis and investigates the corresponding structural features, optical (refractive index, transmittance, optical bandgap) and dielectric properties of the films. They also compared the results with data on relative precursors reported in the literatures. I think that the results are interesting and overall can recommend for publication. The following points should be addressed before publication.

1.    Some figures contain non-explained abbreviations, like Figure 5.

2.     SEM images in Figure 4 are too blurry to make sense.

3.    The data presented in the plot should have error bar.

Reviewer 3 Report

This manuscript employed a phenyl derivative of hexamethyldisilazane – bis(trimethylsilyl)phenylamine as a single-source precursor for SiCN films by plasma-enhanced chemical vapor deposition (PECVD). The SiCN film was characterized by its chemical structure, optical and dielectric properties. However, to improve the current manuscript, I believe the following comments should be addressed:

Comments 1): The introduction provides sufficient background. However, references were not cited for page 2 lines 58-76 properly.

Comments 2): A typo is found by the reviewer. Page 1 Line 19: “Elipsometry”; The academic writing needs to be precise: on page 1 line 22, the author mentioned “defect-free” film, however, on page 7 line 295, “dangling bonds” are attributed to the oxidation of SiCN. To be accepted by Coatings, the reviewer believes that any similar error or typo should be addressed.

Comments 3): The authors used C 1s at 284.6 eV as the reference for binding energy calibration. This is highly unreliable. See Greczynski G, Hultman L. X-ray photoelectron spectroscopy: Towards reliable binding energy referencing. Prog Mater Sci 107, 100591 (2020). The binding energy has to be carefully calibrated to draw any conclusion about the XPS spectra in this work. The ASTM E2108-16 standard (DOI: 10.1520/E2108-16) could be used instead.

Comments 4): The XPS fitting results need to be reconsidered. In Figure 2, the sum and background lines of the fitted curves are not shown. This casts doubt on the reliability of the fitted results and their consistency with the FTIR results. Also, adding a molecular formula may greatly help the public to understand the FTIR and XPS results. The high-resolution scan of O 1s is also missing. The deconvolution results need to be consistent for all the core levels to conclude.

Comments 5): The authors find that the transmittance, band gap, and dielectric constant depend on the composition of the SiCN film because of the deposition temperature for both He and NH3 cases. For the elemental composition (Figure 3), pore volume (Figure 5), and refractive index, consistent trends are shown for both He and NH3 cases as the increase of deposition temperature. However, the band gap shows diverse trends. Please elaborate on the origin of this inconsistency. 

Round 2

Reviewer 1 Report

The authors have revised the manuscript accordingly. 

Reviewer 3 Report

The authors have addressed the comments. The manuscript qualifies for acceptance.